# Responsible Carbon Resource Management through Input-Oriented Cap and Trade (IOCT)

**Lukas Folkens** [1,2,*] **and Petra Schneider** [2]

1    Department for Economics, Magdeburg-Stendal University of Applied Sciences, 39114 Magdeburg, Germany
2    Department of Water, Environment, Civil Engineering and Safety, Magdeburg-Stendal University of
     Applied Sciences, 39114 Magdeburg, Germany; petra.schneider@h2.de
*    Correspondence: lukas.folkens@h2.de; Tel.: +49-391-886-4650

**Abstract:** Fossil fuels store primary carbon. When they are combusted, $CO_2$ is released into the atmosphere. The accumulation of $CO_2$ in the atmosphere causes the anthropogenic greenhouse gas effect, which has led to the existing climate crisis. Academic literature, international climate deliberations and most domestic climate mitigation plans have so far focused primarily on reducing emissions (output orientation) and have paid little attention to supply-side climate policies. Thus, this study shows that output-oriented literature is heavily overweighted with over 7000 publications compared to input-oriented literature with just 107 publications (equivalent to 1.5% percent). The overall scope of this review article was therefore to identify the gaps of output-oriented mechanisms such as the European Union Emissions Trading Scheme (EU ETS), and to point out how an Input-Oriented Cap and Trade (IOCT) system might overcome those gaps. IOCT refers to limits to the carbon input into the global fossil fuel trading system instead of limiting only the emissions caused by already burned fuel. For this purpose, a global cap on the extraction of coal, gas and oil must firstly be defined. Accordingly, IOCT provides for the allocation of allowances for the extraction, processing and trading of carbon-based products. IOCT is a source-oriented approach that refers to a joint allocation of the resource consumption responsibility to the fossil fuel producer and consumer as well. This review represents a unique, comprehensive and current collection of supply-side literature that can be used as a starting point for further applied research on this topic.

**Keywords:** supply side; input-oriented cap and trade; climate policy; fossil fuel production cuts; fossil fuel externalities; limitation of fossil fuel extraction

## 1. Introduction

The year 2020 marked the fifth anniversary of the Paris climate conference. Already in 2019, Ripple et al. together with more than 11,000 scientists warned that a large part of the climate protection plans based on the Paris Agreement were not ambitious enough to prevent untold human suffering caused by an acute climate emergency [1]. By signing the agreement to limit global warming to well below 2 °C, governments have implicitly agreed to drastically reduce the use of fossil fuels over the coming decades, as they are the predominant contributor to climate change [2]. Although the demand for fossil fuels collapsed in 2020 following the COVID-19 pandemic [2,3], the demand in general continues to rise due to increasing global energy consumption. Countries experienced a strong increase in primary energy demand from fossil fuels, especially those with large domestic reserves of fossil fuels [4]. The Production Gap Report 2020 reveals the disconnection between climate goals and energy production plans, as it shows that the planned fossil fuel production is far too high to meet climate targets. For limiting global warming to 2 degrees, the planned production rate by 2030 is about 50 percent too high, and for limiting it to 1.5 degrees, it exceeds the goals by 120 percent [5]. In addition to this, McGlade and Ekins [6] concluded in a 2015 study that about 80 percent of coal reserves, about 50 percent

of natural gas reserves and about 33 percent of petroleum reserves should remain unused in the ground from 2010 to 2050 in order to meet the 2-degree climate goal. A recent study by Welsby et al. [7] updated the numbers relative to 2018 known reserves. According to this study, 58 percent of oil, 59 percent of natural gas and even 89 percent of coal should remain unused by 2050. Based on these findings, Piggot et al. [2] asked why there is still no comprehensive global strategy for phasing out fossil fuel production. Zakkour et al. [8] also noted that supply-side climate policies have been used sparsely, even though there is growing interest in approaches that can limit and end fossil fuel production. Although some authors such as Muttitt and Kartha [9] addressed this issue and outlined how a socially equitable limit on fossil fuel extraction can be realized within climate limits, this review article reveals that most of the academic literature as well as international climate deliberations and most domestic climate mitigation plans have so far paid little attention to supply-side climate policies.

Fossil fuels are carbonated non-renewable resources that are not recyclable after use for fuel purposes. Its combustion releases greenhouse gases (GHG), especially carbon dioxide ($CO_2$). Here, the release of 1 kg of $CO_2$ corresponds quite exactly to the combustion of 3664 kg of carbon (C) [10]. GHG, which according to Rockström et al. already exceed planetary boundaries [11], are the main drivers of the climate crises, which is why their reduction has been given special attention in the international climate debate over the decades. As an example, the emissions trading scheme of the European Union (EU ETS) can be mentioned. Introduced in 2005, the EU ETS is based on $CO_2$ quotas, which are reduced in fixed time steps (cap) and emission permits (allowances), which are traded on designed markets developed for this purpose (trade). Approximately 45 percent of all EU greenhouse gas emissions are regulated by the EU ETS, since only large plants and certain sectors (energy industry, heavy industry, etc.) are covered [12]. Small and very small plants are not involved, which also includes cars and heating systems. This in turn requires individual national regulations, resulting in a patchwork of different measures and laws, which in their entirety should lead to an improvement of the climate situation.

In addition to the output-oriented view (downstream), which focuses on the allocation of emissions, etc., an input-oriented (upstream) approach is also possible as a carbon limiting approach [13]. As early as 1931, Hotelling [14] stated that the extraction of non-renewable resources is economically driven and is only practiced when revenue is granted. He had already considered market limitations, because the extraction would be phased out if there is no economic benefit. Ecological economists such as Daly took up on this and stated that depletion is spatially more concentrated than pollution, which is why the main controls should be at the input end. The resource prices thus increased on the supply-side will force greater efficiency at all upstream stages of production and indirectly limit pollution. Thus, as early as the 1970s, Daly and other ecological economists suggested a cap-auction-trade system for the depletion of basic resources, especially fossil fuels as well as an ecological tax reform [15,16]. According to Erickson et al. (2018) [17], the shift from limiting fossil fuel consumption to limiting fossil fuel production is the next big step in climate policy. According to the laws of chemistry, measures to reduce demand can only curb the greenhouse effect if they also succeed in reducing the supply of carbon [18]. The main purpose of this paper, therefore, is to examine the state of the science in the field of supply-side approaches in the form of a literature review. Particular emphasis is given to cap and trade mechanisms. Initially, this will be limited to fossil fuel production. Particular attention is paid to the questions of:

- why supply-side mechanisms have been taken into account to a very limited extent so far;
- which key aspects must be considered to ensure a successful input-oriented cap and trade mechanism;
- and what measures are necessary to achieve a comprehensive fossil fuel extraction within climate limits [9].

The overall scope of the paper is further to identify the characteristics of the existing mechanisms as well as their gaps, and to point out how an Input-Oriented Cap and Trade (IOCT) system might overcome those gaps. As Figure 1 illustrates, IOCT refers to limits to the carbon input into the global fossil fuel trading system instead of limiting only the emissions caused by already burned fuel. It is a source-oriented approach that refers to a joint allocation of the resource consumption responsibility to the fossil fuel producer and consumer as well. Furthermore, Figure 1 shows IOCT in contrast to conventional output-oriented approaches, here using the EU ETS as an example.

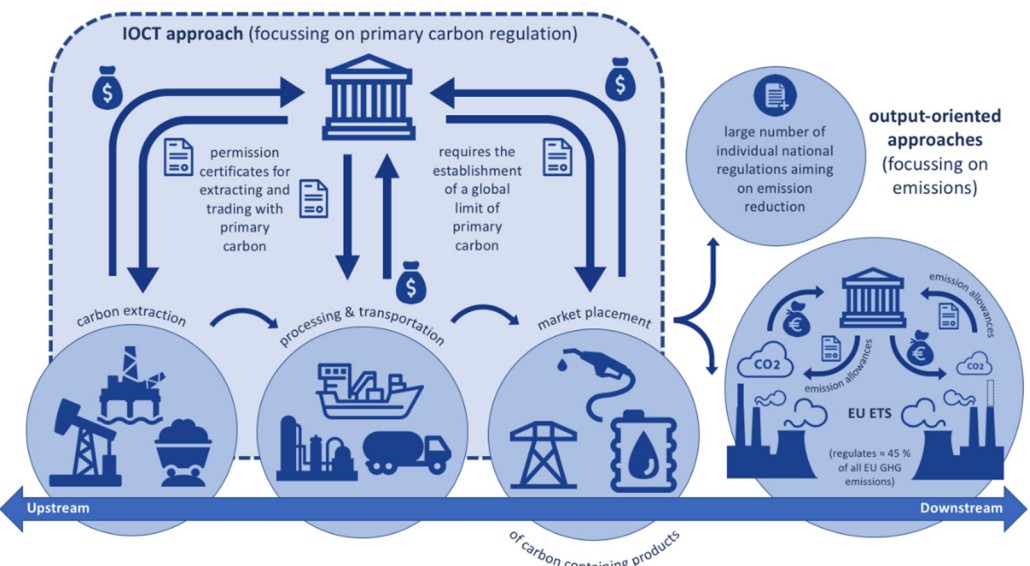

**Figure 1.** Scheme input- vs. output-oriented cap and trade (own illustration).

Recognizing the need to align their climate mitigation plans with their energy production strategies, a small but growing number of governments are beginning to introduce new forms of "supply-side" climate policies [5,17]. For instance, Tudela [19] described the obstacles and opportunities for moratoria on oil and gas exploration or extraction in Latin America and the Caribbean. Furthermore, in addition to the EU ETS, a national emissions trading system based on the Fuel Emissions Trading Act was introduced in Germany in 2020 [20]. This obliges the distributors of fuels to acquire pollution rights in the form of certificates (upstream emissions trading). In other words, they pay for the emissions that result from the subsequent combustion of the fuels. As Piggot et al. (2020) [2] stated and as it is described above, academic literature has lagged behind this policy shift. To determine the state of the science in this research field, a comprehensive literature review was conducted. The methodology used is described in the following section. In addition to the methodological approach, the databases, key words as well as the inclusion and exclusion criteria used are explained in detail. Following up, the results of the study are presented and scientifically discussed in the next section. Thereby, the analyzed literature is compared with each other with regard to defined criteria and its relevance to the research topic. With reference to the outlined research questions, the advantages, barriers and key aspects of IOCT, as well as general aspects for a comprehensive fossil fuel extraction within climate limits are addressed and discussed. In the conclusion the key findings of the study are summarized and placed in an overall context.

## 2. Materials and Methods

In order to analyze our research questions, qualitative research was conducted in form of a systematic literature review [21], starting with a database search of peer-reviewed articles and grey literature using Google Scholar in July 2021 with an update in January 2022 and Scopus in April 2022. Grey literature was considered in the beginning of the

review because IOCT is not yet a very present topic in the scientific community and it should serve for a screening if there are further considerable approaches. Harzing and Alakangas (2016) [22], who did a longitudinal and cross-disciplinary comparison of the three major bibliometric databases Scopus, the Web of Science and Google Scholar, used numerous studies to show that the latter provides a broader coverage than Scopus and the Web of Science. Because the number of publications in Google Scholar is much higher for most disciplines, we restricted ourselves using only this database for our literature review in the first place. We later extended the database search to Scopus, which did not lead to significant further findings. However, it was also pointed out that quite a few of the papers found by Google Scholar are so-called "stray citations", where minor variations in referencing result in duplicate entries for the same paper. Further databases have been verified, such as Environmental Data Explorer, World Input-Output Database, World Development Indicators Database, International Energy Agency Database, Centre for International Earth Science Information Network and ScienceDirect. However, it must be stated that the number of papers addressing IOCT is still very limited, as the majority of the literature is focusing on the conventional cap and trade mechanisms.

Only articles in the English language were considered for the review. No study period was specified in the search. As the overview of our methodological approach in Figure 2 illustrates, we used the following terms in our key word search: "supply side" AND "cap and trade" AND "fossil fuel production" AND "paris agreement", which led to 107 overall results in Google Scholar and just one result in Scopus. Emphasis was put on the status of the carbon flow and consumption systems, and how they might be aligned with the requirements of the Paris Agreement to reduce emissions from fossil fuels. Furthermore, the question was put in the focus during the literature research why supply-side mechanisms have not been taken into account so far as regulating mechanism even they seem to provide much more effective control mechanisms than the conventional cap and trade approaches. The scope was to identify potential IOCT barriers and eventually come up with proposals how to overcome them.

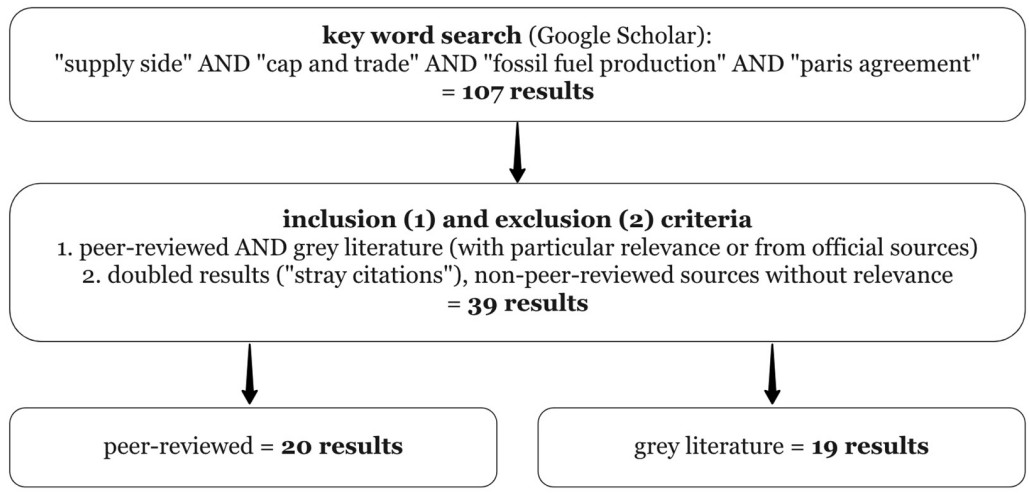

**Figure 2.** Methodological approach (own illustration).

The terms used in key word search influence the entire work to a considerable extent and must therefore be subjected to critical reflection. The challenge here is to keep the search request neither too open nor too narrow. IOCT, at its core, describes a supply-side cap and trade system, so the search terms "supply side" and "cap and trade" were obvious. The combustion of fossil fuels is the main driver of the climate crisis. Since IOCT focuses on the supply side, the term "fossil fuel production" was used. As the approach of the review was to present supply-side mechanisms that are consistent with the Paris climate goals, the term "Paris agreement" was also added. Here, it should be critically noted that the search without "Paris agreement" resulted in 255 hits (as of 30.09.2021). Although this is a larger

number than the actual search, it does not change the basic statement that output-oriented literature heavily outweighs input-oriented literature (Figure 3).

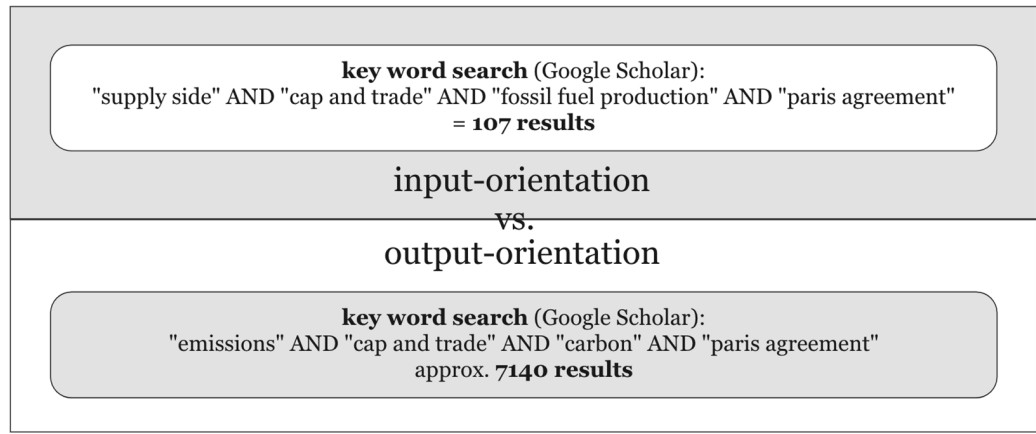

**Figure 3.** Comparison key word search input- vs. output-orientation (own illustration).

In the next step, we defined inclusion and exclusion criteria to filter relevant literature for our research questions. First of all, we excluded all doubled results ("stray citations") as well as non-peer-reviewed sources with no particular relevance. All peer-reviewed papers as well as grey literature with particular relevance and/or from valid official international sources were included. Inclusion criteria are referring to direct relevant information on the research questions mentioned in Section 1. As grey literature with particular relevance were considered papers that fulfilled particular search criteria, that is information on input-oriented (supply-side) fossil fuel handling, and/or information on fossil fuel production cuts. In this regard, publications of official bodies such as the UN have also been considered, as well as scientific documents that have undergone another type of review, such as PhD thesis works. Exclusion criteria refer to non-transparent sources. Scope of this approach was to screen which type of information is available on "input-oriented fossil fuel handling", if practical implementation examples are already available and which lessons can be learned from them. The same applies to the search item "fossil fuel production cuts". The research question should be supported with the information if producers are practicing fossil fuel extraction cuts and if yes, which implications resulted from this approach.

From the initial 107, a total of 39 results remained (36 percent), which can be divided in peer-reviewed (20 hits) and grey literature (19 hits). These 39 results from Google Scholar plus one additional paper from Scopus constituted the research subject for the review and were analyzed with regard to the research questions. In addition to a quantitative analysis, which provides valuable insights into a certain research field [23], literature can be assessed with regard to qualitative criteria (methodology, focus, etc.). If a paper was deemed relevant, the bibliography was also searched for further related literature (snowball technique) [24]. Furthermore, relevant journals were also searched for pertinent Special Issues. In order to structure the totality of the results and to be able to establish references between the sources, we used the software tool Miro.

In addition to comparing qualitative elements such as the main focus of the papers or the methods used, we defined search topics for the remaining results, such as "input-oriented", "fossil fuel production cuts" or "implementation barriers". The entire text of the 40 publications were scanned according to the topics mentioned. Here, the decisive factor was whether the topics were treated in a meaningful way and not whether the exact term occurred. That means that for "input-oriented", for instance, the text refers directly to one or more input-oriented mechanisms. An analogous approach was taken for the remaining two topics. Afterwards we ranked the papers according to their overall relevance, with A meaning high relevance and C meaning low relevance. The assessment of the sources was supported by an intensive scientific discussion using the Miro board in order to identify

interdisciplinary and transdisciplinary interlinkages and particularly connect the ecological, social and economic dimension.

## 3. Results and Discussion

### 3.1. Overview on the Bibliographic Results

Table 1 lists the 39 papers elicited according to the key word search in Google Scholar described in Section 2, with none older than 2016. These are not sorted alphabetically but according to the relevance criteria of Google Scholar search engine. At the end of the list, the paper is additionally mentioned, which was found via the database search in Scopus.

**Table 1.** Results of the Systematic Literature Review.

| Publication | Peer−Reviewed? | Includes "Input−Oriented" | Includes "Fossil Fuel Production Cuts" | Implementation Barriers | Overall Relevance |
|---|---|---|---|---|---|
| Lazarus et al., 2018 [25] | + | + | + | + | A |
| Green et al., 2018 [26] | + | + | + | − | A |
| Gaulin et al., 2020 [27] | + | + | + | − | A |
| Harrison, 2018 [28] | + | − | − | − | C |
| Mitchell−Larson et al., 2020 [29] | − | + | + | + | B |
| Piggot, 2018 [30] | + | + | + | + | A |
| Rempel et al., 2021 [31] | + | − | − | + | B |
| Nicolas et al., 2021 [32] | − | + | + | − | B |
| Mutua, 2019 [33] | − | + | + | − | A |
| Eaton, 2021 [34] | + | + | + | − | A |
| Jenkins et al., 2021 [35] | + | + | + | + | A |
| Linquiti et al., 2016 [36] | + | + | − | + | A |
| Somerville, 2020 [37] | + | + | − | + | A |
| Rissman et al., 2020 [38] | + | + | − | + | B |
| Mendelevitch, 2016 [39] | − | + | + | + | A |
| Armstrong, 2020 [40] | + | + | + | + | A |
| Krane, 2017 [41] | (+) | − | − | + | B |
| Wilde et al., 2017 [42] | − | + | − | + | A |
| Green, 2017 [43] | (+) | − | − | − | B |
| Bernasconi, 2021 [44] | − | + | − | − | C |
| Iacobuță et al., 2021 [45] | + | + | − | − | C |
| Gard−Murray, 2021 [46] | − | + | − | − | C |
| Moz−Christofoletti et al., 2021 [47] | + | − | − | − | C |
| Buck, 2021 [48] | − | + | + | − | B |
| Boyce, 2019 [49] | − | + | − | − | B–C |
| Peszko et al., 2020 [50] | − | + | − | − | B |
| Gupta, 2021 [51] | + | − | − | + | C |
| Baldwin et al., 2020 [52] | + | + | − | + | B–C |
| van der Ploeg, 2020 [53] | + | − | − | − | B–C |

**Table 1.** *Cont.*

| Publication | Peer−Reviewed? | Includes "Input−Oriented" | Includes "Fossil Fuel Production Cuts" | Implementation Barriers | Overall Relevance |
|---|---|---|---|---|---|
| Noisecat, 2021 [54] | − | + | + | + | A |
| Okoh, 2021 [55] | + | + | − | + | B |
| Abraham−Dukuma, 2021 [56] | − | + | + | − | C |
| MacLean, 2019 [57] | − | + | − | + | A |
| Oberthür et al., 2019 [58] | − | + | − | − | B–C |
| Drudi et al., 2021 [59] | − | + | − | + | C |
| Albrecht, 2021 [60] | − | + | − | − | B |
| Moss et al., 2020 [61] | − | + | − | − | C |
| Bolton, 2020 [62] | − | − | − | + | C |
| Sæther, 2016 [63] | − | + | + | − | B–C |
| Li et al., 2022 [64] | + | + | + | + | A |
| | 21 | 32 | 16 | 20 | A = 15 |

+ match, − no match, (+) no impact factor, A high relevance, B medium relevance, C low relevance.

The literature listed illustrates that there are feasible economic, political and social approaches to reduce fossil fuel production consistent with climate goals. In addition to the database search, we searched journals of particular relevance for suitable Special Issues. This is how we became aware of the Special Issue "Curbing Fossil Fuel Supply to Achieve Climate Goals" of Climate Policy (20:8, 2020). The editorial by Piggot et al. (2020) [2] has been cited several times in this article, as it deals with comparable research questions, such as barriers to, and opportunities for, supply-side approaches. Furthermore, some of the papers listed above such as Green et al. (2018) [26] or Gaulin et al. (2020) [27] are part of the mentioned Special Issue.

To first get an overview of supply-side initiatives, reference can be made to Gaulin and Le Billon (2020) [27], who presented the first global database of supply-side climate initiatives seeking to constrain fossil fuel production. The database contains over 1300 initiatives. Green and Denniss (2018) [26] also broke down supply-side policies and measures in their climate policy toolkit dividing between restrictive policies such as fossil fuel subsidy reductions, supply taxes, production quotas and bans or moratoria on extraction and supportive policies such as direct government provisions of low-carbon infrastructure as well as renewable energy feed-in-tariffs. A comprehensive taxonomy, including beside others cap and trade for production rights or offsets for leaving assets in the ground in addition to those already mentioned, is also presented by Lazarus and van Asselt (2018) [25]. With reference to Metcalf and Weisbach (2009) [65] they suggest that the most efficient point of regulation for a cap and trade system would be upstream at the point of fossil fuel extraction. Building up on this, they cite Harstad (2012) [66], who suggests that trading for rights to extract fossils fuels could have advantages over trading for rights to emit GHG, as well as Collier and Venables (2015) [67], who propose that aforementioned should be coupled with strategies for phasing out coal production. Overall, we have succeeded in filtering out literature that deals with input-oriented approaches at its core, whereas the overwhelming majority of academic literature, as will be shown, deals with demand-side policies. 32 of the 39 sources indicated in Table 1 refer directly to one or more input-oriented approaches. According to this, IOCT is placed in the center of the following considerations, whereby other input-oriented mechanisms are also mentioned repeatedly. IOCT would mean that the extraction and trading of fossil fuels would be linked to permission certificates. Importing or extracting fossil carbon would thus become subject to permits and federal authorities would issue

annually decreasing import or extraction quotas (cap), which they would auction. Green and Denniss (2018) [26] among others, e.g., [39] concluded that the evaluation of IOCT and most other supply-side instruments has largely been neglected in the literature. Properly applied, IOCT provides political, economic, environmental and social advantages, as the following section reveals.

### 3.2. Advantages of IOCT (and Other Supply-Side Mechanisms)

Compared to emissions trading, which starts where $CO_2$ emissions leave the economy, IOCT has the major advantage that extractions and imports occur at relatively few points, so this measure would be easy to implement, with lower administrative and transaction costs [25,26,68] and a maximum coverage as well as higher allocative efficiency [25,68] coming along with the reduction of bureaucratic burden. The amount of extracted carbon can be accounted for at a fraction of the effort required to quantify any $CO_2$ emissions [68]. Further economic advantages as a higher abatement certainty, comprehensiveness within-sector coverage, mitigation of infrastructure "lock-in" risks or mitigation of the "green paradox" are presented by Green and Denniss (2018) [26]. Mutua (2019) [33] discussed unilateral actions for contributing to climate change mitigation by limiting own oil extraction in Norway. The results support that supply-side policies belong in the optimal mix and it is cost effective for Norway. The author underlined that a combination of demand-side and supply-side climate measures is preferred, and future research should look at the adaptations of the taxation system and the adjustments needed to restrain exploration activities. This opinion was also supported by Sæther (2016) [63].

Lazarus et al. (2018) [25] argued that supply-side policies and actions will tend to slow investment in fossil fuel production and trade infrastructure, may increase moral pressure and public support for climate action. Restrictive supply-side policies have been used effectively in other contexts, tobacco being a prominent example. Just as in other markets, reducing the supply of oil and gas by limiting the development of those resources' production and transmission, infrastructure will tend to increase their prices and, in turn, reduce their demand [57]. Moreover, Lazarus et al. (2018) [25] criticized that demand-side policies were not ambitious enough and not applied widely enough, from removing fossil fuel subsidies, to taxing production, as well as to retiring assets.

In order to foster IOCT, awareness has been created through the concept of "Fossil Fuel non-proliferation treaty" (NPT) by Newell and Simms (2020) [69]. The authors highlight the need for international agreements and law to effectively and fairly leave large swathes of remaining fossil fuels in the ground. The NPT has a three-pillar structure: non-proliferation, disarmament and peaceful use. Nicolas et al. (2021) [32] and Iacobuţă et al. (2021) [45] substantially supported the approach of this initiative. Moreover, Harstad (2012) [66] and Asheim et al. (2019) [70] proposed already comparable initiatives in the past. The SDGs are one of the instruments to create awareness for the environmental dimension of energy supply including the aspect of intergenerational cost justice [46]. The global energy market does not include in its pricing mechanism, like many other markets, untaxed negative externalities [71], for instance caused by climate extremes [51]. Untaxed negative externalities are a main cause for intergenerational injustice. Exemplarily, Gupta (2021) [51] mentioned USD 500 billion in losses from 2015–2019 only in the U.S due to climate extremes. In this context, Drudi et al. (2021) [59] mentioned also the relation to the impacts of climate extremes on the functionality of the supply side. There is an undue profit and a constraint on carbon emitting activities because these costs for society are not paid leading to a continuous output of carbon emissions. Even economists such as Green (2017) [43] mentioned the role of negative externalities in relation with GHG emissions, the public awareness for them is typically underdeveloped. There might be a variety of reasons, however missing transparency in global production and supply chains is considered as one of the key issues by the authors. Referring to Hotelling [14] and his statement that the extraction of non-renewable resources is economically driven and only practiced when revenue is granted, the conclusions implicate that there is an intrinsic relation between ongoing extraction

and negative externalities. Those externalities might typically result from environmental and social cost that are usually not considered in the mining life cycle even though first attempts have been performed in recent years to mitigate this gap, for instance, by the International Council on Mining and Metals (ICMM) [72]. IOCT would internalize some of the externalities addressed by harmonizing private sector and societal marginal costs.

### 3.3. Barriers for IOCT

Even though input-oriented policies got a growing momentum [23,27], Figure 3 illustrates the preponderance of demand-side cap and trade literature by showing how much academic literature exists relative to input-oriented ones.

For this comparison, we replaced the search terms "supply side" and "fossil fuel production" with "emissions" and "carbon". Based on the findings, we examined the literature to search why supply-side mechanisms are so underrepresented and where the barriers to their implementation are. As Table 1 shows, 20 out of the 39 papers reviewed discussed implementation barriers of IOCT.

Lazarus et al. (2018) [25] identified existing power relations between governments and fossil fuel industries, as a main implementation barrier. Stepping back from fossil fuel production is therefore challenging for most fossil fuel producing countries, because of "carbon entanglement" (Gurría (2013) [73]). Fossil fuel rich countries have an economic interest in ensuring that existing stocks are brought to market, as they themselves benefit from, e.g., royalty payments or taxes. Another type of financial incentives are subsidies. According to de Bruin et al. (2019) [74], a subsidy is classified as potentially environmentally damaging if it is likely to cause incentivize behavior that could be damaging to the environment irrespective of its importance for other policy purposes. As they stated [74], the common fossil fuel subsidies concern sectoral fossil fuel subsidies to lower the cost of production or to secure the supply of certain commodities for air and land transportation as well as electricity and peat production sectors. Moreover, there are commodity related subsidies to lower the retail prices of certain commodities, including car and truck diesel, kerosene and fuel oil [74]. de Bruin et al. (2019) [74] propose to combine the removal of sectoral fossil fuel subsidies with a carbon tax to increase the carbon emission reduction efficiency. Examples for producer subsidies are tax credits on exploration and production, direct payments per unit or low rates on leasing of nationally owned land [75,76]. Gençsü et al. (2020) [77] showed for the European Union that governments provide EUR 21 billion per year of support for fossil fuel production. Erickson et al. (2017) [78] pointed out that nearly half of new, yet-to-be-developed oil investments in the USA depend on subsidies to be profitable. According to the OECD/IEA [79], in G20 countries, fossil fuel subsidies amounted to more than USD 159 billion in 2020, compared to USD 162 billion in 2010. Other types of mining subsidies are for instance the missing taxation of water use for mining operations. According to Mendelevitch (2016) [39], taking the example of steam coal, the eight major producers that benefit from subsidies are the USA, China, India, Australia, South Africa, Indonesia, Russia and Poland. The subsidies level varies significantly between 0.1 USD/t in Poland and 3.4 USD/t for coal from the U.S. Powder River Basin (PRB). Depending on the producer this corresponds to less than 1% of production cost for Poland and South Africa, up to 34% for PRB coal [39]. Subsidies impact the quantity of fuels supplied to markets, so it is anticipated that their elimination could ultimately reduce fuel consumption and thus $CO_2$ emissions [80]. Moreover, subsidies and other production-enhancing measures have led to the fact that real energy prices for fossil fuels have remained low in relation to income development, which slows down the further development of clean energy and efficiency technologies, as they require long periods to earn a return [38].

Not only there is a lack of exit strategies for fossil fuel producing and exporting countries, but according to Armstrong (2020) [40], the sensitive issue of losing out on a significant source of revenue, particularly for some of the world's poorest countries is needed to be addressed. The aspect of acceptance therefore plays an important role. For capacity reasons, this issue is underrepresented in this paper and would have to be

investigated in more detail in subsequent studies. Moreover, Linquiti et al. (2016) [36] supposed that many of the fossil fuel actors such as mining companies and operators are in a position to thwart policy changes because of the expected losses of their business field. The authors wonder if there is a way to compensate them for their losses, so as to facilitate the process of decarbonization and propose a benefit transfer from winner to looser market participants. However, Linquiti et al. (2016) [36] also recognized that such an approach on international scale is impractical. Linquiti et al. (2016) [36] also pointed out that if only some resource owners limit fossil fuel production, the production would likely migrate to other unconstrained resource owners, thereby enriching them and also undermining the environmental benefit. It should be emphasized that IOCT can only be meaningfully implemented as part of a global effort involving all relevant mass flows. What would happen if, for example, the European Union decided to implement IOCT and to link the import and production of fossil fuels to the possession of tradeable allowances? Those companies that want to trade with fossil fuels within the EU or bring them into circulation would need the corresponding permits. Demand would subsequently decrease in Europe but increase in other countries with a lack of regulatory measures because the global price of fossil fuels is lower in this case. Thus, unless there is a global strategy to phase out fossil fuel production, they will continue to be extracted and subsequently consumed. In economic terms, this phenomenon is called market clearing.

Critical words have been issued by Krane (2017) [41] who pointed out that type and intensity of climate risk differs greatly among the three forms of fossil fuels, as well as between countries in the developing and developed world. The author summarized an increased risk for coal industry and compared to this a reduced risk for oil businesses, due to its lack of substitutes [41]. Those findings also correlate with the fact that the coal industry is already phased out in some industrialized countries, such as Germany. The coal phase-out was most recently endorsed by over 40 signatory nations at COP26, including major coal users such as Poland, Vietnam and Chile, but not the U.S., India or China [81]. However, the necessary decarbonization of many sectors (e.g., mobility, industry) is forcing these to electrify, resulting in an enormous explosion in demand for electrical energy. Base-load capable power generation sources are very difficult to identify, except for coal and nuclear power generation, which are considered problematic for other reasons. Regardless of which energy mix scenarios are favored in the future, they will all lead to a drastic increase in prices, which in turn will require the acceptance of consumers. This is opposed by the fact that goal 7 of the UN Sustainable Development Goals (SDGs) aims at affordable and clean energy by 2030. In present, only a few countries at the global scale rely completely on clean energy headed by Albania, Paraguay and Norway, and followed by Iceland, Tajikistan, Costa Rica, Nicaragua and New Zealand [82]. While the first mentioned countries to a large extent rely on hydropower, Iceland also uses geothermal, wind and solar power, and New Zealand focuses on wind and solar power. Additionally, Costa Rica and Nicaragua produce clean electricity from hydro, geothermal, solar and wind. However, Bolton [62] reminded that there might be barriers as well that could stand in the way of smooth development of renewable energy capacity as alternative energy source. In contrast, other countries are relying to a large extent on fossil fuels, or are dependent on them, such as Singapore (98 percent), Australia (93 percent), South Africa (91 percent), as well as Luxembourg and The Netherlands (90 percent). While Australia, South Africa and The Netherlands own substantial reserves of fossil resources, Singapore and Luxembourg rely on fossil fuel imports. Other counties, such as Germany, are fostering the energy transition to renewables. As a matter of fact, for many countries, fossil fuels are still the priority choice due to lower energy cost, energy security from own sources, but also because of high infrastructure and societal transition cost. As Peszko et al. (2020) [50] underlined, fossil fuel-dependent countries face financial, fiscal and macro-structural risks for a low-carbon transition away from carbon-intensive fuels and value chains. For those countries according to Albrecht (2021) [60] collaborative strategies for diversification and decentralization are needed. A recent publication by Okoh (2021) [55] illustrated what a

future after low-carbon transition of a fossil fuel rich country could look like, in this case Nigeria, counting on biofuels. Another strategy for alternative fuel sources that is followed up recently particularly in the European Union is green hydrogen.

In addition, some of the existing fossil fuel-based infrastructure will have to be abandoned or replaced [52,53,83]. Rempel and Gupta, 2021 [83], even assume that COVID-19 might be accelerating the fossil fuel asset stranding process. Moreover, fossil fuel associated labor force need to be provided with opportunities for a just transition of their income options [54]. However, a distinction must be made between the various forms of fossil fuels. While natural gas networks are also suitable for a post-fossil energy supply (e.g., hydrogen) [84,85], coal infrastructure is unsuitable in this context. Mitchell-Larson et al. (2020) [29] mentioned IOCT implementation barriers, such as no well-defined policy framework that effectively incentivizes the permanent storage of carbon in non-atmospheric pools, and carbon sequestration solutions are at various stages. Therefore, in 2021, Mitchell-Larson and Allen (2021) [86], proposed a carbon takeback / storage obligation approach. Another main implementation barrier according to Lazarus et al. (2018) [25] are changes in individual as well as social-structural behavior. This is taken up by Piggot (2018) [30], with particular reference to social and political barriers to mobilization focused on restricting fossil fuel supply. Last but not least, Abraham-Dukuma (2021) [56] identified corruption, political instability and weak democratic institutions in many resource-rich countries as IOCT barriers.

### 3.4. General Aspects for a Comprehensive Fossil Fuel Extraction within Climate Limits

Kartha et al. (2016) [87] mentioned two main perspectives on equity when it comes to the extraction of fossil fuels: 1. extraction as pollution, 2. extraction as development. These two opposing views have also become clear in the preceding remarks. The first essentially states that fossil fuel extraction is the ultimate source of fossil carbon emissions and therefore suppliers should be treated as polluters. The second perspective is that for fossil fuel producing and exporting countries, these resources are an important contributor to development [87]. The question of supply-side restrictions arises in this area of tension. In this paper, priority was given to the former perspective in order to be able to clarify the IOCT approach. In this context Eaton (2021) [34] summarized that decades of demand-side green capital advocacy and policy approaches, the crisis of global heating is only accelerating. As Johnsson et al. (2019) [4] concluded, immediate and disruptive changes to the use of fossil fuels are required to achieve the climate goals, which implies only two principal mitigation options for fossil-fuel-rich economies: (1) leave fossil fuels in the ground; and (2) apply carbon capture and storage (CCS) technologies. While the latter option is excluded from the current consideration, the first one got special attention, as 16 of 39 papers evaluated in our article contain the topic of "fossil fuel production cuts". As Gaulin and Le Billon (2020) [27] stated, there is a clear imperative to keep a large proportion of fossil fuel reserves underground to be in line with the Paris Agreement. Several countries such as France, Belize, Denmark, New Zealand, Ireland, Spain and Germany have already implemented bans on fossil fuels exploration or production [88]. Somerville (2020) [37] also addressed the urgent need of action directed at delegitimizing fossil fuels and stopping their extraction. He proposed, among other measures, a significant tax on carbon embedded in imports.

Rissman et al. (2020) [38] assembled technical and policy interventions, both on the supply-side and on the demand-side, that can achieve net zero industrial emissions in the timeframe 2050 to 2070. The authors underlined that environmental core policies should be supported by labeling and government procurement of low-carbon products, data collection and disclosure requirements, and recycling incentives. From our point of view, these are also relevant aspects as governments should not only regulate but also take over a role model. Buck (2021) [48] underlined that a net zero goal is ambitious, however, this goal offers balance and stability as well as ambiguity which offers a wide range of ideas and practices to be developed. In this regard, Bernasconi (2021) [44] proposed a net zero

approach that accounts for nature-based solution as a kind of carbon buffer system that credits supply-side emissions reductions efforts. East Germany might serve as a role model here, where the main activities of coal mining were closed since 1990, restoration is taking place resulting in a large-scale green infrastructure. However, it must also be recognized that the regional transformation in East Germany led to an enormous loss of economic activity and a large number of people had to leave the region for that reason. Mendelevitch (2016) [39] went even beyond simple policy intervention and discussed a moratorium on new coalmines as a supply-side climate policy, that was also initiated by the President of Kiribati. However, there is only very little literature on this subject, particularly Erickson and Lazarus (2016) [89] and Finighan (2016) [90].

Wilde et al. 2017 [42] summarized that the oil taxation literature has recognized that a royalty could lead to marginal fields not being developed, and that this might be considered a weakness of a royalty compared to more neutral profits and excess profits-based taxes. Due to the fact that there is also a concern that a high oil royalty in one country might lead to companies choosing to invest in the development of fields in lower tax/royalty jurisdictions, Wilde et al. 2017 [42] proposed a global oil royalty that is implemented at the same rate by all countries. One of the open questions regarding a global oil royalty is, which body would administrate and enforce it.

*3.5. Key Aspects for Successful IOCT*

After explaining the barriers to input-oriented approaches in general and IOCT in particular, as well as general aspects for comprehensive fossil fuel extraction within climate boundaries, this section focuses on what key aspects need to be considered for a potential successful implementation of IOCT. Resource extraction is an optimization process, however not only from economic, but also ecological point of view. It must be noted that fossil carbon resources are non-renewable and therefore cannot be managed "sustainably" but according to the principles of "responsible mining", taking into account that their stock is limited. As this paper makes clear, supply-side climate policies do not yet play a major role neither in the literature nor the public narrative. Consequently, the opportunities and potentials of IOCT would first have to be brought into sharper focus scientifically and finally introduced into the public discourse. Factors such as opening windows of political opportunity and compelling frameworks that call citizens to action may also play a role in this context [30,91].

The supply chain of $CO_2$ emissions [92] starts where the carbon is extracted from the ground. In the existing climate discourse, however, mostly those measures are discussed that lie at the end of this supply chain, i.e., mostly where the $CO_2$ leaves the economic system. Following on from this disparity, the introduction of an IOCT system implies a transition toward supply-side constraints. An implementation strategy will require the adaptation of international raw material agreements taking into account environmental stewardship including the social relations with the local communities. The first step in that direction would be to remove fossil fuel mining subsidies. This can be considered one of the key aspects to ensure a successful input-oriented cap and trade mechanism. Removing all open and hidden fossil fuel subsidies will already lead to higher fossil fuel prices and foster the transition to renewable energies. Effective implementation also requires the establishment of a global limit for primary carbon, for which permission certificates for extracting and trading are then allocated and gradually lowered over defined time periods. It is important that the burdens associated with this transition are shared between the global north and the global south [40]. The main perspectives of Kartha et al. (2016) [87], described at the beginning of the previous section, suggest that transition approaches such as IOCT must place social and economic justice at the center of its struggles if it is to have broad-based appeal [34]. Dealing with the producers of fossil fuels plays a key role in this regard [27]. Implementation must not be carried out in isolation from them. While the global setting of caps on primary carbon seems unrealistic, it is essential in order to address uneven adoption and prevent the relocation of production [27]. However, there is still a

risk that individual actors would dominate the market and manipulate prices. Interest groups could buy quotas at auction and sell the imported energy sources exclusively to their members. This risk would need to be kept in mind when implementing IOCT. One possible way would be that no player would be allowed to bid for more than a certain percentage of the total quota. On the other hand, Jenkins et al. (2021) [35] proposed carbon takeback obligation (CTBO) combined with demand-side mitigation and concluded that it provides a low-risk pathway to net-zero emissions by 2050.

## 4. Conclusions

This review represents a comprehensive and current collection of supply-side literature that can be used as a starting point for further applied research on this topic. The study shows that output-oriented literature is heavily overweighted with over 7000 publications compared to input-oriented literature with just 107 publications (equivalent to 1.5% percent). Dealing with the question of how to ensure a sustainable energy supply for soon-to-be 10 billion people is becoming one of the central tasks of the current century. This radical change will not succeed without a drastic reduction in the use of fossil fuels, as they are the predominant contributors to the climate emergency, which will require more than ever the inclusion of supply-side measures. Thus, the study will create awareness for the need, the options and the possibilities of the IOCT approach, having a more critical look on conventional cap and trade mechanisms and understanding of the limited success of conventional cap and trade mechanisms. The necessary measures to achieve a comprehensive fossil fuel extraction within climate limits can be considered as follows:

- Create a global understanding on phasing out fossil fuels through an input-oriented mechanism, ideally by putting it into an international agreement, for instance an UN convention.
- Preparation of the practical phasing out of fossil fuels through elimination of subsidies or similar hidden support in the existing fossil fuel extraction processes.
- Continuation of the establishment of green and renewable energy sources to ensure alternative energy sources, including upgrading grids and infrastructure if necessary.
- Continuation of the development and implementation of energy efficient technologies on all levels in order to reduce the energy and fuel consumption.
- Mitigate potential resource and energy scarcity conflicts through a participation and dialogue process as well as financial support if necessary.
- Implement the renewable energy production systems on a global scale.

**Author Contributions:** The concept of the article was developed at the University of Applied Sciences Magdeburg-Stendal, section Ecological Engineering, by L.F. and P.S. The introduction as well as the results, methodology, discussion and conclusion sections were written by both authors, whereby L.F. conducted the original draft preparation and P.S. the review and editing. The formal analyses, investigation and visualization were performed by both authors. P.S. did the validation of the literature included as well as the funding acquisition through her guest editor role. All sections have been commented by the co-authors and include their views. All authors have read and agreed to the published version of the manuscript.

**Funding:** This research received no external funding.

**Institutional Review Board Statement:** Not applicable.

**Informed Consent Statement:** Not applicable.

**Data Availability Statement:** Not applicable. All data sources are cited in the reference list.

**Conflicts of Interest:** The authors declare no conflict of interest.

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
