# Peer review of "Responsible Carbon Resource Management through Input-Oriented Cap and Trade (IOCT)"

_sustainability, doi:10.3390/su14095503_

Round 1

Reviewer 1 Report

The study reviewed emissions-cap trading that helps to limit carbon emissions. The study is well written and achieved the stated objectives, however, I have some comments on a paper that needs to be incorporated, i.e.,

i) Introduction: Add possible research question(s), and clearly stated research objective(s).

ii) Add more current literature up to 2022 in the manuscript.

iii) Add contribution of the study.

iv) Results should be discussed in a more critical manner and in line with the earlier studies.

v) Long-term policy implications should be added accordingly. 

Author Response

Thank you for your comments. Based on these and other suggestions, major changes were made to the manuscript. 

point-by-point response:

i) was revised and updated accordingly;

ii) Three recent papers have been added: Li et al., COP26 and Hielscher et al.;

iii) was revised and updated accordingly;

iv) was revised and updated accordingly;

v) they have been added to the conclusion section

Reviewer 2 Report

This is a very intriguing review paper focusing on the supply-side climate policy and IOCT.

Author Response

Thank you for your comment. Major changes were made to the manuscript.

Reviewer 3 Report

This scientific article that we have appraised is a high quality work.

It is well written, clear, precise, relevant in its form and especially in terms of its demonstration. The methodology is clearly presented, the demonstration is of quality and allows a valid answer to the questions asked. The figures are numerous and really support the demonstration.

The bibliographical references are numerous, recent, in short it is a very relevant scientific analysis.

This excellent quality research work deserves to be validated as it is.

Author Response

Thank you for your comments. Major changes were made to the manuscript.

Reviewer 4 Report

The subject of the article is interesting and worth describing. However, the method of implementation requires correction. In the Introduction, the authors presented an introduction to the subject. The Introduction section is deficient. It does not contain all the necessary elements. The main purpose of the research was not clearly defined. There are research questions in the Introduction section, which I assess positively. At the end there should be information about the content of each section.
The layout of the work is basically correct, but slight adjustments need to be made. I have already listed what should be in section 1 Introduction. The Materials and Methods section is correct. Includes all required items. The authors describe the sources of the materials and the methods used. However, I have doubts about the methodology itself. Choosing only the Google Scholar database, and omitting Scopus and Web of Science (WoS) is, in my opinion, a mistake. The authors try to explain themselves. However, if they were to decide on one database, it should be Scopus or Web of Science. The authors themselves admit that the Google Scholar database contains errors, not all articles are scientific. There are no such doubts in the case of Scopus and WoS. The choice of the Google Scholar database is therefore a big methodological error.
The obtained search results are not satisfactory. Ultimately, 39 articles were analyzed. A little. Perhaps the problem was the too short period analyzed.
The Results section actually contains a scientific discussion. It should be called Discussion. On the other hand, the Discussion section has also been created and it is not significantly different from the Result section. Review papers should contain syntheses or generalizations or to summarize the state of the art in some field. This was not the case in the reviewed article. The authors did not use any methods that could allow for generalizations.
Conclusion section is incomplete. The conclusions can be bulleted. One must certainly refer to the research questions posed. Have these questions been answered.
Reading the article, I have the impression that it is a good literature review for the research part of the article. Meanwhile, such research is missing here. I understand that it was supposed to be a review, but in its current form, the article does not meet the requirements for articles of the review type.

Author Response

Thank you for your comments. Based on these and other suggestions, major changes were made to the manuscript.

point-by-point response:

  • Introduction: The information about the content of each section was added.
  • Database: Based on this comment, a database search was started in Scopus, which resulted in a single paper for the same key words (Google Scholar: 107 papers). This result underlines the statements in chapter 2. Nevertheless, explanations for the use of Scopus were added accordingly. The fact that in the end, only 39 articles remained underlines the core statement of the review paper, because it shows how underrepresented input-oriented compared to output-oriented literature is. Furthermore, we stated at the beginning of the second paragraph in chapter 2 that no restriction was made with regard to the study period.
  • Results and Discussion: The Discussion section has been integrated into sections 2 and 3.
  • Conclusions: The Conclusions section was revised and updated.
  • Last paragraph: The scope and meaning of the comment of the reviewer is unclear to the authors. The manuscript respects Scientific Best Practice in terms of the structure, the research methodology and the results. The topic is completely new, and a review paper on IOCT does not exist yet in the scientific literature.

Round 2

Reviewer 4 Report

The main remark concerned the use of articles from Google Scholar only. The authors stated in a revised version that they also used the Scopus database. However, I have my doubts that it has happened to the full extent. I checked the corrected article. The changes have been marked in a different color. Only one item was added in Table 1: Li et al, 2022 [64]. I doubt these databases are so consistent. This doubt still remains. I leave it for consideration for the Authors and for the editorial staff of Sustainability.
Of course, a few references have been added to the literature when describing the results.
The Results and Discussion section has been merged, which can be assessed positively. This way, there will be no more doubts.

Author Response

For the response to the review report please find the attached document.
